# High-Definition 3D Exoscope-Assisted Barbed Pharyngoplasty for OSAS and Snoring: Better Than Live

**DOI:** 10.3390/healthcare11040596

**Published:** 2023-02-16

**Authors:** Manuele Casale, Antonio Moffa, Michelangelo Pierri, Peter Baptista, Lucrezia Giorgi

**Affiliations:** 1Integrated Therapies in Otolaryngology, Fondazione Policlinico Universitario Campus Bio-Medico, 00128 Rome, Italy; 2School of Medicine, Campus Bio-Medico University, 00128 Rome, Italy; 3Department of Otorhinolaryngology, Clinica Universidad de Navarra, 31007 Pamplona, Spain; 4ENT Department, Al Zahra Private Hospital Dubai, Dubai 23614, United Arab Emirates; 5Unit of Measurements and Biomedical Instrumentation, Department of Engineering, Campus Bio-Medico University of Rome, 00128 Rome, Italy

**Keywords:** sleep apnea, barbed pharyngoplasty, 3D exoscope, high definition

## Abstract

Recently, a high-definition 3D exoscope (VITOM), a new magnification system that provides a 3D image of the surgical field, has been introduced. This study aims to describe the first use of VITOM 3D technology in Barbed Pharyngoplasty (BP) for Obstructive Sleep Apnea (OSA). VITOM 3D technology is used to support visualization during BP in a male patient affected by severe OSA with a circular palatal collapse pattern at drug-induced sleep endoscopy. During the surgical procedure, this approach markedly improves the visualization of the surgical field through anatomic details of the oral cavity, facilitating surgical dissection and enhancing the teaching environment. It allows for a better involvement and more interactions during the surgery, as scrubbed and assistant nurses can see the surgical field and anticipate the surgeon’s choice of instrument. VITOM 3D technology, by combining a telescope with a standard endoscope, has been successfully used in various surgical disciplines and could be very useful, especially in teaching hospitals. VITOM 3D can guarantee “a real immersive” surgical experience for all participants in the operating room. Economic and efficacy studies would be conducted to support the use of a VITOM-3D exoscope in common clinical practice.

## 1. Introduction

In recent years, the use of endoscopic procedures in the ENT field has gained a lot of popularity. In this context, optical magnification has become an essential tool in ENT surgery. This approach is based on a specially designed scope connected to a high-definition digital camera and an HD video monitor [1]. The VITOM 3D exoscope (Karl Storz Endoscopy, Tuttlingen, Germany) is an emerging and interesting technology that aims to provide enhanced visualization of the surgical field. Indeed, as it allows magnification to everyone in the operating theatre, it is useful for sharing the different steps of surgical procedures, documenting uncommon cases, and training young surgeons. The VITOM 3D exoscope has already been widely used in different types of ENT surgery, as reported by a recent systematic review [2]. As an alternative to the conventional microscope or loupe magnification, this 3D exoscope was used in many different surgical procedures, such as in head and neck reconstruction [3], microvascular anastomosis [4], rhinoplasty [5], lacrimal surgery [6] and ear surgery [7]. However, to date, there are no studies on the use of VITOM 3D for oropharyngeal Obstructive Sleep Apnea (OSA) and snoring surgery. In the last few years, the introduction of Barbed Sutures (BSs) has greatly changed the surgical treatment of patients suffering from snoring and OSA with retropalatal collapse and vibration. Many surgeons have discovered the advantages and unique properties of BSs, which has facilitated the revision and improvement of the most popular surgical pharyngoplasty techniques [8]. Among the different types of BP, there is the Alianza barbed pharyngoplasty technique, which is indicated to correct the circular palatal collapse caused by the combination of the anteroposterior and lateral components for OSA patients and snorers [9]. In this type of surgery, the main drawback is that surgical maneuvers are limited within a very small surgical field, such as the mouth (not infrequently, OSA patients have small mouth openings); therefore, it is crucial to magnify and see in detail the soft palate and all the fibro-muscular structures while maintaining complete anatomic fidelity. Moreover, not all operators can see the surgical field in the same way, and so they cannot adequately help the first operator. This study aims to describe the first use of VITOM 3D technology in Alianza pharyngoplasty in a male patient affected by severe OSA with a circular palatal collapse pattern.

## 2. Materials and Methods

VITOM 3D is a 0° telescope with a diameter of 10 mm and a length of 10 cm. The camera head is mounted on the proximal end of the telescope, and the illumination is provided by a cold light fountain Xenon 300.

The VITOM 3D exoscope was used to perform Alianza Barbed Pharyngoplasty. The VITOM 3D exoscope is a video system that combines 4K resolution view and 3D technology, displaying images on a flat screen. 3D vision is possible using dedicated glasses. The 3D camera, provided with a magnification power of 8–30 and a depth of field between 7 mm and 44 mm, is held by a supporting arm and supports a focal distance of 20–50 cm. It is attached to a ceiling-mounted supporting arm and placed in front of and above the patient’s mouth, about 25 cm away from the operating field. Because of its position outside the patient’s body, it is called an exoscope. Its limited dimensions and positioning are convenient for the operator, who is free to choose to stay in a sitting or standing position, behind the patients’ head (Figure 1). The VITOM 3D exoscope is controlled by an intuitive sterile joystick, which can be used to modify magnification, focus, and other video settings. When the exoscope is positioned, details of lesions are anatomically observed on a HD monitor. Additionally, the nurse or the second operator can rotate, with fine movements of the scope, the supporting device in the three planes of space so that the operating field is always centered on the screen.

## 3. Results

A 55-year-old male patient with moderate OSA was referred to our institution (Unit of Integrated Therapies in Otolaryngology, Campus Bio-Medico University Hospital Foundation, Rome, Italy). Before the surgery, the patient underwent an evaluation that included the Home Sleep Apnea Test (HSAT) performed using the Watch-PAT device, a sleep history assessment including the Epworth Sleepiness Scale (ESS), a severity of snoring assessment with a numeric Visual Analogue Scale (VAS) and an awake endoscopic evaluation.

He underwent an inferior turbinate reduction and Drug-Induced Sleep Endoscopy (DISE), which showed a complete concentric collapse at the velum. The patient refused to use the CPAP, and the case was presented at the OSA board. After discussion of the case, it was concluded that the patient should be treated with Alianza pharyngoplasty under general anesthesia.

All the surgical steps of Alianza were performed using a VITOM 3D exoscope. The use of an exoscope enhanced the visualization of the palatopharyngeal muscle, thus facilitating the dissection from the superior constrictor muscle. Moreover, it allowed us to identify the ptherigomandibolar raphe, to perfectly enter the BS into the hole where we exited from and to perform a rapid and meticulous hemostasis while avoiding damage to contiguous anatomical structures (Figure 2).

Using the VITOM 3D, there was no additional time spent operating. Moreover, no major or minor complications were described.

After the surgery, the patient was advised to take a liquid diet for the first 24 h and a soft diet for the next 2 weeks. A normal diet can be started after 2 weeks. We recommend Chlorhexidine mouthwashes after each meal during the first postoperative week. We give 1 g of paracetamol three times a day for 5–7 days for analgesia, and we add Ibuprofen for breakthrough pain. In addition, we give Amoxicillin/clavulanate (875/125 mg), administrated every 12 h for 5 days.

Postoperatively, an ENT examination, snoring VAS and ESS assessments, and PSG were performed 6 months after surgery. We recorded a significant improvement in the main PSG values (AHI from 32.5 to 8.5 episodes/hour, ODI from 29.5 to 7.0 episodes/hour) and in the snoring VAS (from 8.5 to 3.0).

## 4. Discussion

Since a rapid development in technology occurred in the last decade, new diagnostic and surgical instruments in the ENT field are currently used.

The VITOM-3D exoscope system (Karl Storz GmbH, Tuttlingen, Germany) has the advantages of being less expensive and smaller than traditional microscopes and of thus adding a superior image quality to a small footprint and better ergonomics [10].

This study describes the first use of VITOM 3D technology in Alianza barbed pharyngoplasty in a male patient affected by severe OSA with circular palatal collapse pattern. In the literature, only one study has used this device for a transoral snore surgery, but they reported a cadaver simulation [11].

VITOM 3D resulted in being very useful in the intrapharyngeal OSA and snoring surgery. First of all, the VITOM system allows for a depth of field ranging from 3.5 cm to 10 cm and a magnification of 12-30x, which is comparable to the ENT operative microscope. In this way, VITOM 3D allows for a full stereopsis at magnification and the addition of color filters (Clara/Chroma/Spectra) highlighting the interface between different tissue types, such as blood vessels, nerves, muscles, mucosa and the submucosa layer. It was very useful to identify all the critical structures including the palatopharyngeal muscle and pterygomandibular raphe, for instance for hemostasis purposes [12]. Usually, VITOM 3D is placed 25 cm away from the target area, providing the surgeon with a wide range of movement without interfering or risking a collision with the camera. Another important advantage of the exoscope is that all operators can share the same view of the surgical field. In many surgical steps, the second operator cannot see the operating field very well and fails to help, especially for the steps performed in the deeper portion of the field [13]. In this way, a better involvement and more interactions have been shown during the surgery.

This aspect is crucial for teaching new surgical techniques to trainees and young surgeons. Comparing 3D-exoscope-assisted surgery with loupe magnification, the latter can also provide a good magnification of the surgical field, but it is limited by the fixed loupe’s magnification and both less and more difficult sharing of the surgery [14]. The possibility of placing the exoscope and the 3D monitor in virtually all possible positions with different orientations allows the surgeon to always adopt a comfortable position. Indeed, the first surgeon is not forced to assume a specific position. The VITOM system provides a considerable advantage in allowing a neutral cervical spine position and natural spinal posture for the surgeon. In 72% of surgeons who experience neck and/or back pain, the main cause is maintaining a static body position with extended periods with neck flexion [15].

With the use of VITOM 3D, the surgeons could maintain a comfortable position, and all the observers in the operating room could share a clear and magnified view of the intraoral surgical field with the first surgeon. Moreover, the exoscope enabled the precise placement of the BS at the soft palate and the lateral wall in order to improve the tissue stiffness and muscle translocation and to reduce the risk of suture dehiscence. In addition, it is important to note that the use of 3D VITOM caused no additional operating-room time compared with the standard procedure, and no major or minor complications were observed. This result is confirmed by previous studies on the use of the VITOM 3D in ear and lateral-skull-base surgical procedures compared to traditional methods [7,16]. Actually, it has been reported that the use of the VITOM 3D has shortened the operative time, which can correlate with a cost-effective benefit of the exoscope over a modern surgical microscope [16,17].

Finally, the learners can better understand all the anatomical structures involved in and different steps of palate-pharyngeal surgery [18]. Moreover, scrubbed and assistant nurses can directly observe the surgical field and anticipate the surgeon’s choice of instrument.

The role of VITOM 3D during the training of young surgeons has been previously investigated by Molteni et al. [19] in a microvascular-anastomosis training setting. However, even if they observed an absolute improvement of mean scores after almost all the steps, no significant differences were seen between the operating microscope and the VITOM 3D. Since statistical significance was not reached, the authors showed a noninferiority of the 3D exoscope compared with the operating microscope for the training of surgeons in microsurgery. Indeed, the advantages that the VITOM 3D can bring to the training of young surgeons and improvement of their performance need to be investigated with further studies.

The Video 3D setting can guarantee “a real immersive” surgical experience for all the participants in the operating room. Additionally, the VITOM 3D system could be useful not only for oropharyngeal surgery but also for tongue-base surgery in cases of tongue-base obstruction.

## 5. Limitations

The main limitation of this report was the use of the VITOM 3D exoscope on a single patient and the lack of objectively quantifiable outcome measures; thus, our findings are anecdotal and have limited generalizability. In addition, the subject was evaluated at 6 months after surgery, whereas it will be useful to evaluate outcomes over a longer follow-up period. However, with an increasing number of cases treated at our hospital and extended postsurgical observation times, we would be able to gain further insight into the long-term success rate.

## 6. Conclusions

VITOM 3D markedly improves the visualization of oropharynx anatomical details and provides a careful surgical dissection of surgical planes with a real 3D-4K magnification, improving the understanding of surgical technique and enhancing the teaching environment. In conclusion, it could represent a useful tool for barbed pharyngoplasty. Moreover, it allows the surgeon to operate from a comfortable posture with the possibility of sharing the main surgical steps with the entire operative team. Economic evaluation studies should be performed on the use of the VITOM-3D exoscope, considering not only the cost of this new tool but also the operative time and great advantages related to it.

## Figures and Tables

**Figure 1 healthcare-11-00596-f001:**
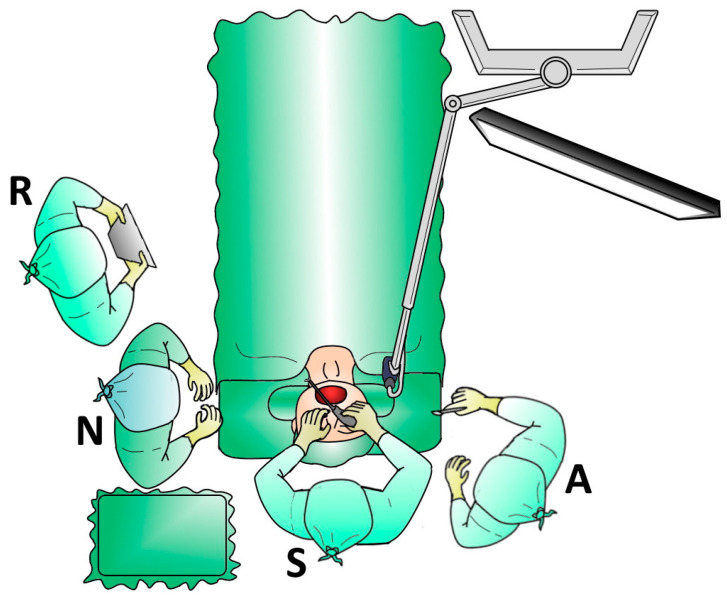
Operating theater setting for barbed pharyngoplasty using the 4K three-dimensional exoscope system (VITOM 3D): Surgeon (S), Assistant (A), Nurse (N), Resident (R).

**Figure 2 healthcare-11-00596-f002:**
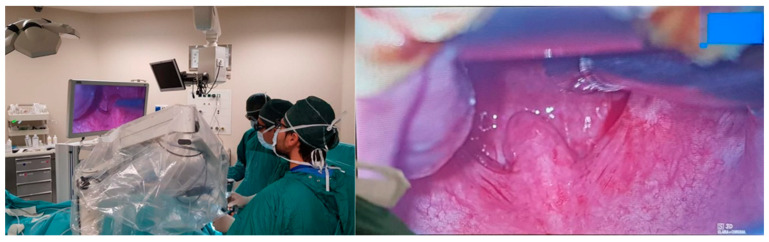
Intraoperative view of oropharynx.

## Data Availability

The datasets generated during and/or analyzed during the current study are available from the corresponding author on reasonable request.

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
