# Peer review of "High-Definition 3D Exoscope-Assisted Barbed Pharyngoplasty for OSAS and Snoring: Better Than Live"

_healthcare, 2023, doi:10.3390/healthcare11040596_

Round 1

Reviewer 1 Report

The 3D exoscope introduced in this report is a kind of observation system that allows surgeons, assistants and nurses to see clearly the structures in oropharynx. It is positioned in the outer field away from the operation field, rendering the disadvantages such as obstacles from surgeon’s hands and instruments introduced into oropharynx during the operation. Therefore, the 3D endoscope is much more useful for the operation purpose than the 3D exscope described in the present manuscript.

It seems that the author is very confident about the usefulness of 3D exoscope in the BP pharyngoplasty for OSA and snoring, however, this no solid evidence exampling the application of the exoscope in the surgical procedure of pharyngoplasty in OSA and snoring patients. The only picture provided here is the glance or overview of the surface structure of the oropharynx, observed on the screen of the monitor. It does not necessarily mean that 3D exscope introduced in this manuscript can satisfactorily fulfil the task of bettering BP pharyngoplasty in the treatment of OSA and snoring.

Author Response

  1. The 3D exoscope introduced in this report is a kind of observation system that allows surgeons, assistants and nurses to see clearly the structures in oropharynx. It is positioned in the outer field away from the operation field, rendering the disadvantages such as obstacles from surgeon’s hands and instruments introduced into oropharynx during the operation. Therefore, the 3D endoscope is much more useful for the operation purpose than the 3D exscope described in the present manuscript.

Thank you very much for your comment. It is not necessary  to use the endoscope instead of the exoscope because VITOM 3D allows us to visualize all the operative field with a correct depth. As recently described in other surgical procedures it can be useful. Iwami K, Watanabe T, Osuka K, Ogawa T, Miyachi S, Fujimoto Y. Combined Exoscopic and Endoscopic Technique for Craniofacial Resection. Curr Oncol. 2021 Oct 4;28(5):3945-3958. doi: 10.3390/curroncol28050336. PMID: 34677254; PMCID: PMC8535086. Furthermore, the endoscope, given its shape, would hinder the surgeon in performing the main surgical step. For these reasons, the use of the endoscope to perform barbed pharyngoplasty procedures is not indicated.

  • It seems that the author is very confident about the usefulness of 3D exoscope in the BP pharyngoplasty for OSA and snoring, however, this no solid evidence exampling the application of the exoscope in the surgical procedure of pharyngoplasty in OSA and snoring patients. The only picture provided here is the glance or overview of the surface structure of the oropharynx, observed on the screen of the monitor. It does not necessarily mean that 3D exscope introduced in this manuscript can satisfactorily fulfil the task of bettering BP pharyngoplasty in the treatment of OSA and snoring.

Thank you very much for your comment. It is the first report describing the application of the VITOM 3D to barbed pharyngoplasty surgical procedure. It is a non-essential tool but it can be a useful tool to support the surgeon and to show the main surgical steps.  Obviously further studies will be needed to confirm this result.

Reviewer 2 Report

the results section should be completely revised.

you wrote it like a summary conclusion

you have to present certain items (e.g: time of operation, -----etc.)

after revision of result section, the discussion section should be changed accordingly.

Author Response

  1. The results section should be completely revised, you wrote it like a summary conclusion, you have to present certain items (e.g: time of operation, -----etc.). After revision of result section, the discussion section should be changed accordingly.

We agree with the reviewer and we modified the results section: “We hereby the case of a 55 years old male patient with moderate OSA who was referred to our institution. He underwent inferior turbinate reduction and DISE which showed complete concentric collapse at the velum. After the discussion of the case at the OSA board, the patient that refused CPAP and He was treated with Alianza pharyn-goplasty under general anesthesia.Preoperative assessment included Home Sleep Test (HST) using the Watch-PAT de-vice endoscopic evaluation, sleep history assessment including the Epworth Sleepiness Scale (ESS), and severity of snoring assessed with a numeric Visual Analogue Scale (VAS). All the surgical step were performed using VITOM 3D exoscope. The use of exoscope enhanced the visualization of the palatopharyngeal muscle, thus facilitating the dissec-tion from the superior constrictor muscle. Moreover, it allows to identify the ptherigoman-dibolar raphe and to enter with BS perfectly in the hole where we went out and to perform a rapid and meticulous hemostasis avoiding damage to contiguous anatomical structures (Fig.2). Vitom 3D was very useful in all BPs steps, in particular to dissect the palatopharyngeal muscle to the superior constrictor muscle, to identify the ptherigomandibolar raphe and to enter with BP perfectly in the hole where we went out and to perform a rapid and meticulous hemostasis (Fig.2). Second operator, anesthesiologist, nurses, residents, and students can have the same perspective of the surgeons and, through 3D technology, perceive well the depth of the surgical field. We did not have additional operating room time. No major or minor complication were described. Patient is advised to take liquid diet for the first 24 h and soft diet for the next 2 weeks. Normal diet can be started after 2 weeks. We recommend Chlorhexidine mouthwashes after each meal during the first postoperative week. We give 1 g of paracetamol, three times a day for 5–7 days for analgesia and we add Ibuprofen for breakthrough pain. In addition, we give Amoxicillin/clavulanate (875/125 mg) given every 12 h for 5 days. Post-operatively, ENT examination, snoring VAS and PSS assessments and PSG were performed at least 6 months after surgery. We recorded a significant improvement in the main PSG values (AHI from 32.5 to 8.5 episodes/hour, ODI from 29.5 to 7.00 episodes/hour) and at Snoring VAS (from 8.5 to 3).”

We modified the discussion section and we added “The VITOM-3D exoscopic system (Karl Storz GmbH, Tuttlingen, Germany) is less expensive and smaller than microscopes and possesses the additional advantage of better image quality, ergonomics and a small footprint. Iwami K, Watanabe T, Osuka K, Ogawa T, Miyachi S, Fujimoto Y. Combined Exoscopic and Endoscopic Technique for Craniofacial Resection. Curr Oncol. 2021 Oct 4;28(5):3945-3958. doi: 10.3390/curroncol28050336. PMID: 34677254; PMCID: PMC8535086.”

….With the use of VITOM 3D, the surgeons could maintain a comfortable position, and all the observers in the operating room could share a clear and magnified view of the intraoral surgical field with the first surgeon. Moreover, the exoscope enabled the precise placement of barbed sutures at the soft palate and the lateral wall in order to improve the tissue stiffness and muscle translocation and to reduce the risk of suture dehiscence. We did not have additional operating room time. No major or minor complication were described.”….

…”Limitations

The main limitation of this report was the use of a single patient, and the lack of quantifiable outcome measures; thus, our findings are anecdotal and have limited generalizability. The small number of cases and the short follow-up period are also major limitations. Nevertheless, with an increasing number of cases treated at our hospital, and extended postsurgical observation times, we would be able to gain additional insight into the long-term success rate.”

Reviewer 3 Report

Dear Authors, thank you for your paper. It is very interesting and innovative as focusing on the application of new medical device in oro-pharyngeal surgery. However, I suggest to improve the paper with images of the surgical procedure, as you report only the pre-operative one. As so the paper is only a report about a new medical device. Thank again for your paper.

Author Response

  1. Dear Authors, thank you for your paper. It is very interesting and innovative as focusing on the application of new medical device in oro-pharyngeal surgery. However, I suggest to improve the paper with images of the surgical procedure, as you report only the pre-operative one. As so the paper is only a report about a new medical device. Thank again for your paper.

Thank you for the comment. Unfortunately, we only have these images. We hope it will be useful for publication.

Round 2

Reviewer 1 Report

Although only one case of application of the described exoscope is presented, it still can not completely resolve my cercerns and anwser the questions raised.  The image provied is only the primary observation of the appearance of oral pharynx at the beginning of the pharyngopalsty. No representative intraoperative images  were presented. In may opinion, intraoperative video recording the application of the exoscope during the whole procedure have to be provided. I still believe that the instrument described here is suitable for open surgery rather than transoral pharyngeal surgeries.

Author Response

Thank you again for your comments. Unfortunately, for this study we do not have the video of the whole procedure but only the images shown. In future studies we will show a detailed video of the procedure. 

Reviewer 2 Report

accepted

Author Response

Thank you very much. 

Reviewer 3 Report

Thank you for revised paper. All changes have been done as required. Thank you for you work.

Author Response

Thank you very much.